# Psychometric Results of a New Patient-Reported Outcome Measure for Uveal Melanoma Post-Brachytherapy Treatment: The PROM-UM

**DOI:** 10.3390/cancers15164142

**Published:** 2023-08-17

**Authors:** Ezekiel Weis, Jing Jiang, Alison H. Skalet, Carol L. Shields, R. Trafford Crump

**Affiliations:** 1Department of Surgery, Cumming School of Medicine, University of Calgary, Calgary, AB T2N 4N1, Canada; 2Departments of Ophthalmology, Casey Eye Institute, Oregon Health & Science University, Portland, OR 97239, USA; 3Ocular Oncology Service, Wills Eye Hospital, Thomas Jefferson University, Philadelphia, PA 19107, USA; carolshields@gmail.com

**Keywords:** uveal neoplasms, brachytherapy, patient-reported outcome measures, psychometrics

## Abstract

**Simple Summary:**

Based on previous work, we designed a new questionnaire that asks patients to report how they feel after undergoing plaque radiation treatment for a certain type of eye cancer. This study aimed to test that the questionnaire measured what it was intended to measure. The questionnaire was given to 439 patients from three clinics in Canada and the United States of America. The results indicated that the questionnaire measures three domains: symptoms and their impact on the patient, the patient’s worry, and the patient’s discomfort. The questionnaire is easy to use and reliable. The questionnaire can help doctors manage patient care after the treatment.

**Abstract:**

The objective of this study was to evaluate the psychometric properties of a new patient-reported outcome instrument intended for use with patients who have undergone brachytherapy for uveal melanoma (PROM-UM). Classical test theory and item response theory were used to evaluate the performance of individual items and domains. A convenience sample of 439 participants who had undergone brachytherapy for uveal melanoma from one of three North American ocular oncology treatment centers were included in this cross-sectional study. Exploratory factor analysis identified three domains which were labelled “Symptom Impairment”, “Worry”, and “Discomfort”. The acceptability of the instrument was supported by little missing data (range = 0.00–1.14%) and low maximum endorsement (range = 0.00–1.82%). Item-total (range = 0.68–0.85) and inter-item (range = 0.74–0.80) correlations indicated acceptable reliability. Discrimination and difficulty were assessed using item response theory. Items in all three domains indicated moderate to very high discrimination (range = 1.00–4.10). Two items in the Symptom Impairment domain were too difficult to measure. Response ranges in the other two domains demonstrated acceptable difficulty. These results from the study indicate that this new patient-reported outcome instrument can be used with patients treated with brachytherapy for uveal melanoma. Providers could use this instrument to help inform post-treatment management.

## 1. Introduction

Uveal melanoma can be a devastating diagnosis with fatal consequences and severe vision loss. It is the most common primary intraocular malignancy in adults, with an incidence rate of approximately 5.1 cases per million people in the United States [1]. Uveal melanoma has a disease-specific mortality rate of 45% at 15 years [2]. 

Historically, enucleation (i.e., removing the eye) is the most common and only treatment for uveal melanoma. The emotional challenges associated with losing an eye and the functional challenges of monocular vision can be very impactful for patients. Advances in radiation and surgical techniques have led to brachytherapy as a treatment alternative. With brachytherapy, not only is the saved eye more aesthetically appealing to patients, but also the functioning vision from that eye may be retained in a significant proportion of patients, with no decrease in overall survival [3]. However, brachytherapy for uveal melanoma is also associated with more medical management challenges than enucleation. By saving the eye, patients may experience changes in their visual acuity, pain, and distress related to the risk of local recurrence [4]. 

Being able to measure patient symptom severity after brachytherapy can therefore help inform post-treatment management. One way that symptom severity can be measured is through patient-reported outcomes (PROs). PROs are validated instruments that ask respondents to self-report their symptoms and functional impairments [5]. When collected systematically, PROs can be used to track changes in symptom severity and impairment over time. In addition, utilising a standardised and validated tool can provide an important method of evaluating the effect of changes made to the treatment and patient support system.

The European Organisation for Research and Treatment of Cancer (EORTC) has developed the OPT-30, a PRO instrument specifically for use with patients diagnosed with eye cancer, including uveal melanoma [6]. The OPT-30 is a 30-item instrument that asks symptom-specific questions related to vision and cancer. The OPT-30 is intended to be paired with a 30-item generic quality-of-life instrument (EORTC’s CTC-30) that asks all cancer patients about their physical, social, and emotional functional status; symptoms of fatigue, nausea, vomiting, and pain; and global health status. These instruments, totaling 60 questions, are intended to be used in clinical trials. 

While the OPT-30 is a very comprehensive, validated instrument, its clinical application may be limited. A qualitative study of the instrument observed that the OPT-30 could have better questions regarding mental health; the impact of diagnosis and treatment on family; thr burden of treatment on patient and caregivers, including travel and financial commitments; and the impact on work and home life [7]. Moreover, the OPT-30 is likely too long for many clinics to be used at the point of care, especially if paired with the generic CTC-30. A study observed that it took patients, on average, 23 min to complete the OPT-30 alone [7]. 

Thus, there is a need for a more concise PRO instrument for real-world, real-time use with patients diagnosed with uveal melanoma and treated using brachytherapy. Based on previous validation work, we designed an initial version of the PROM-UM in English. The purpose of this study is to validate a measure of patient-reported outcomes following brachytherapy for the treatment of uveal melanoma.

## 2. Materials and Methods

### 2.1. Development of an Initial Patient-Reported Outcome Instrument

Prior research developed an initial PRO instrument based on focus groups with patients diagnosed with uveal melanoma and those who underwent brachytherapy [7]. Focus groups were drawn from samples of patients from clinical programs at the University of Alberta in Edmonton, Alberta, Canada; the Casey Eye Institute at the Oregon Health Sciences University in Portland, Oregon, United States; and the Stein Eye Institute at the University of California—Los Angeles, in Los Angeles, California, United States. 

The initial PRO instrument comprised 17 items related to a symptom or concern. Respondents were asked to rank each item based on the severity of their respective impact/experience using a four-point Likert scale: Not at all, A little, Quite a bit, Very much.

### 2.2. Field Testing the Initial Patient-Reported Outcome Instrument

After the development phase was completed, the initial PRO instrument was field tested at three clinical programs: the University of Alberta; the Casey Eye Institute; and the Wills Eye Hospital in Philadelphia, Pennsylvania, United States. A convenience sample of patients who had undergone brachytherapy for their uveal melanoma (different from those recruited for the focus groups) at each centre were asked to complete the PRO using paper and pencil or through an electronic survey system (depending on the center). 

Response data were anonymised and sent to the investigators for analysis. This study’s protocol was approved by the Health Research Ethics Board of Alberta—Cancer Committee. 

### 2.3. Exploratory Analysis

Due to the low number of enucleation patients not allowing sufficient data collection, one item in the initial PRO instrument was removed because it was intended for patients who had undergone enucleation. Responses to the remaining 16 items were converted to a numeric scale ranging from 0 (i.e., for responses = Not at all) to 3 (i.e., for responses = Very much).

An exploratory factor analysis (EFA) was used to identify the number of latent traits/domains [8,9,10]. Factors with an eigenvalue ≥ 1 were retained and considered unique domains. Items with factor loading ≥ 0.30 after a promax rotation and that did not have cross-loading issues were retained. Cross-loading was defined as an item loading ≥ 0.30 on more than one factor, or when the difference between the main factor load and secondary factor load was <0.20 [11]. If a cross-loading issue was observed, we eliminated the items that had this issue one by one, conducted further principal-factor analyses of the correlation matrices of the remaining items, and checked the eigenvalues and factor loadings again. We repeated the process above until all remaining items successfully loaded on only one factor. Items not loading on any factor were eliminated.

Factor scores were calculated by adding the scores of each item within the factor. If an item’s score was missing, it was excluded from the analysis; no imputation was attempted. 

### 2.4. Classical Test Theory Analysis

**Acceptability**. We evaluated acceptability in two ways: missing data percentage and maximum endorsement (MEF). The missing data percentage was calculated as the proportion of non-responses for each item over the total number of participants, ideally less than or equal to 5% [12]. The MEF was calculated as the percentage of items receiving the highest possible score, ideally less than 20% [12].

**Reliability**. Reliability was tested in two ways. First, it was evaluated in an inter-item manner defined as the internal consistency within a factor. This was measured using Cronbach’s Alpha, where a value of >0.70 is considered acceptable. Second, we employed an item-total approach evaluating how well each item within a factor is correlated to the summation of factor score, subtracting the item itself [13]. This correlation is referred to as the “item-total correlation”. It was measured by the Pearson correlation between an individual item and the factor score omitting the item. Item-total correlations greater than 0.30 were considered acceptable [12,14].

### 2.5. Item Response Theory Analysis

**Graded Response Model (GRM)**. For each domain, a GRM was used to estimate the domain’s respective items’ discrimination parameter (α) and difficulty parameter (β). The discrimination parameter represents the slope of the item characteristic curve. It illustrates how well an item distinguishes respondents at different gradients of the latent trait. Discrimination thresholds can be considered: none (α = 0), very low (0.01 ≤ α < 0.34), low (0.35 ≤ α < 0.64), moderate (0.65 ≤ α < 1.34), high (1.35 ≤ α < 1.69), and very high (α ≥ 1.7), where higher discrimination is better.

The difficulty parameter measures how difficult an item is for respondents with the latent trait to provide the appropriate response to. In our instrument, each item had four levels of response, with higher levels indicating more severe symptoms/concern. Difficulty is defined as the point at which a respondent with the latent trait has a 50% chance of responding in the correct category or higher. Difficulty thresholds can be considered: very easy (β ≤ −2), easy (−2 < β < −0.5), medium (−0.5 ≤ β ≤ 0.5), hard (0.5 < β < 2), and very hard (β ≥ 2).

As missing data were a necessary component of our analyses (i.e., acceptability), there was no attempt at imputation. All analyses were completed using Stata SE 18 (StataCorp. 2023. Stata Statistical Software: Release 18. College Station, TX, USA: StataCorp LLC). 

## 3. Results

There were 439 participants sampled in this study from the three ophthalmology practices. The majority of these (57%; n = 248) were from the University of Alberta, followed by the Wills Eye Hospital (33%; n = 143) and the Casey Eye Institute (11%; n = 48).

The initial EFA identified two items that had cross-loading issues. We systematically removed those with cross-loading issues one by one. Further factor analyses were conducted after each elimination. Those items that were eliminated were not put back into the PRO instrument.

The ultimate EFA identified three factors with eigenvalues ≥ 1. As detailed in Table 1, all remaining items loaded on one of these factors with no cross-loading issues. Based on our interpretation of these items, we labelled Factor 1 as the “Symptom Impairment” domain comprising seven items with a domain score range between 0 and 21 (with higher values indicating more severe symptoms). Factor 2 was labelled the “Worry” domain comprising three items and a domain score range between 0 and 9 (with higher values indicating more worry). Factor 3 was labelled the “Discomfort” domain comprising four items and a domain score range between 0 and 12 (with higher values indicating greater discomfort).

### 3.1. Classical Test Theory Results

Table 2 details the results of the acceptability and inter-item and item-total reliability tests. The acceptability of the PRO instrument is supported by low missing data percentage (range = 0.00–1.14%) and low MEF percentage (range = 0.00–1.82%). The inter-item reliability of the PRO instrument is supported by the acceptable Cronbach’s Alpha coefficients of each domain. The item-total correlation ranges for all three domains (0.65–0.72 for Symptom and Impairment; 0.82–0.88 for Worry; 0.66–0.81 for Discomfort) also support the instrument’s reliability.

### 3.2. Item Response Theory Results

The GRM estimates of the discrimination and difficulty parameters are provided in Table 3. For the Symptom Impairment domain, two items could be characterised as having “medium” discrimination, three “high”, and two “very high”. This indicates that all items differentiate between individuals possessing similar levels of the latent trait.

In terms of difficulty, only one item in the Symptom Impairment domain (i.e., Blurry vision) had an even balance between easy and very hard response range. Four items (i.e., Vision loss, Sense of depression, Sense of anxiety, and Driving difficulty) ranged from “medium” to “very hard”. The other two items had response ranges that were “hard” to “very hard”. This indicates that respondents with the latent trait may have had a difficult time scoring the appropriate response.

For the Worry domain, all three items had a “very high” type of discrimination, indicating the ability to differentiate between those with and without the underlying trait. All three items had well-balanced response ranges in terms of their difficulty.

Items in the Discomfort domain ranged from “moderate” (i.e., Grittiness in eye or socket) to “very high” (i.e., Pain in the eye or socket and Pressure in the eye or socket) in terms of their respective discrimination. All items ranged from “moderate” to “very hard” in terms of their difficulty. All items differentiated between individuals possessing similar levels of the latent trait.

## 4. Discussion

Uveal melanoma is the most commonly treated malignancy for most ocular oncologists in Northern America and Europe. The close to 50% long-term survival, the significant visual morbidity with radiation treatment, and the worries of cancer recurrence make the treatment of uveal melanoma challenging. 

Validated PROs can improve our ability to efficiently and effectively evaluate symptoms from the patient’s perspective. When used in real time, they can be completed in the waiting room by the patient while awaiting to see the clinician; key patient concerns can be identified and quickly reviewed during the patient encounter. By utilising standardised and validated measures, real meaning can be attributed to these scores to understand how impactful they are to the patients. As a result of clearly identifying these issues, interventions can be initiated to address the patients’ symptoms and concerns. Evidence supports the use of PROs to improve clinical outcomes in cancer care [15].

Psychometric evaluation of the PROM-UM instrument from three centres in North America was performed on patients who underwent treatment for their uveal melanoma. Factor analysis revealed that the items fell into three domains. These three domains can be categorised into a “Symptom impairment” domain which includes symptoms of poor vision, mental health, and performing regular activities; a “Worry” domain which includes concerns over cancer recurrence and metastasis; and a “Discomfort” domain which includes issues of pain and pressure. 

Classical test theory demonstrated the instrument to be acceptable and reliable. Evaluation using item response theory demonstrated that the response scales for a couple of items in the “Symptom impairment” domain were difficult, but for the “Worry” and “Discomfort” domains the response scales were balanced. This may warrant future research, but the instrument can be used now with the understanding that it may under-represent the severity of some symptoms.

Since many local ocular symptoms, such as those addressed in this instrument in patients undergoing treatment for uveal melanoma, are transient, further research including patients in the first few months after treatment evaluating the responses and proximity to treatment may allow for further understanding of these responses.

The results from this study may be limited in their generalisability given the sampling methods and its size. A random sample of patients treated with brachytherapy for uveal melanoma could generate different results. Having demonstrated the initial psychometric performance of the PROM-UM, we can refine it further through more sophisticated methods in future research. 

## 5. Conclusions

The PROM-UM instrument was developed for use with patients undergoing treatment for uveal melanoma. Due to the rarity of enucleation, psychometric evaluation was only performed on those undergoing radiation treatment in the form of brachytherapy. Three domains including “Symptom impairment”, “Worry”, and “Discomfort” were identified. Classic test theory and item response theory demonstrated acceptable performance in all aspects of the instrument except in the Symptom impairment domain. Further research at varying time points after treatment is recommended to evaluate these results.

## Figures and Tables

**Table 1 cancers-15-04142-t001:** Rotated factor loadings (pattern matrix) and unique variances for the Exploratory Factor Analysis.

Variable	Factor 1	Factor 2	Factor 3
Vision loss	**0.44**	0.13	0.02
Blurry vision	**0.52**	0.06	0.11
Sense of depression	**0.55**	0.32	−0.07
Sense of anxiety	**0.63**	0.28	−0.20
Difficulty driving	**0.51**	0.22	0.13
Daily activities	**0.99**	−0.24	0.09
Inability to work	**0.88**	−0.12	0.05
Cancer recurrence	−0.14	**0.88**	0.06
Cancer spreading to other eye	−0.06	**0.79**	0.14
Cancer spreading to other body parts	−0.10	**0.89**	−0.04
Pressure in eye or socket	0.03	0.04	**0.85**
Pain in the eye or socket	0.05	0.04	**0.81**
Grittiness in eye or socket	0.08	−0.08	**0.58**
Headaches	−0.03	0.10	**0.64**

Bold indicates unique factor loading.

**Table 2 cancers-15-04142-t002:** Results of acceptability and reliability tests, by domain (N = 439).

Domain	Number of Items	Cronbach’s Alpha	Mean Item-Total Correlations (Range)	Missing Data %	MEF %
Symptom impairment	7	0.79	0.68 (0.65–0.72)	1.14	0.00
Worry	3	0.80	0.85 (0.82–0.88)	0.00	1.82
Discomfort	4	0.74	0.76 (0.66–0.81)	0.00	0.00

**Table 3 cancers-15-04142-t003:** Estimated discrimination and difficulty parameters from a Graded Response Model.

Variable	Discrimination Estimate (α)	SE		Difficulty Estimate (β)	SE
**Symptom Impairment Domain**					
Vision loss	1.00	0.13			
			≥1	−0.39	0.12
			≥2	0.95	0.15
			=3	1.99	0.25
Blurry vision	1.21	0.14			
			≥1	−0.58	0.12
			≥2	1.16	0.15
			=3	2.29	0.25
Sense of depression	1.67	0.20			
			≥1	0.28	0.09
			≥2	1.76	0.17
			=3	3.02	0.32
Sense of anxiety	1.59	0.18			
			≥1	0.04	0.09
			≥2	1.47	0.14
			=3	2.44	0.24
Difficulty driving	1.65	0.19			
			≥1	0.13	0.09
			≥2	1.66	0.16
			=3	2.89	0.30
Daily activities	3.91	0.62			
			≥1	0.56	0.07
			≥2	1.60	0.11
			=3	2.12	0.16
Inability to work	3.25	0.45			
			≥1	0.77	0.07
			≥2	1.53	0.11
			=3	1.97	0.15
**Worry Domain**					
Cancer recurrence	4.02	0.81			
			≥1	−0.13	0.06
			≥2	1.00	0.09
			=3	1.64	0.12
Cancer spreading to other eye	2.92	0.41			
			≥1	0.34	0.07
			≥2	1.43	0.11
			=3	2.17	0.17
Cancer spreading to other body parts	2.20	0.23			
			≥1	−0.66	0.09
			≥2	0.78	0.09
			=3	1.56	0.13
**Discomfort Domain**					
Pressure in the eye or socket	4.10	0.89			
			≥1	0.48	0.07
			≥2	1.54	0.12
			=3	2.39	0.20
Grittiness in eye or socket	1.24	0.16			
			≥1	0.29	0.10
			≥2	2.19	0.25
			=3	3.76	0.48
Pain in the eye or socket	3.27	0.56			
			≥1	0.43	0.07
			≥2	1.89	0.14
			=3	2.79	0.26
Headaches	1.49	0.19			
			≥1	0.32	0.09
			≥2	1.99	0.20
			=3	3.21	0.35

## Data Availability

The data presented in this study are available on request from the corresponding author. The data are not publicly available due to potential privacy concerns.

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
