# Peer review of "Psychometric Results of a New Patient-Reported Outcome Measure for Uveal Melanoma Post-Brachytherapy Treatment: The PROM-UM"

_cancers, 2023, doi:10.3390/cancers15164142_

Round 1
Reviewer 1 Report
Thank you for the opportunity to review the paper "Psychometric results of a new patient-reported outcome measure for uveal melanoma post-brachytherapy treatment: The PROM-UM" by Weis et al.
The evaluation of the psychometric properties of assessments is of utmost importance to be able to use them meaningfully in clinical practice. Therefore, I would like to thank the authors for their efforts.
However, I think some concerns could be considered when revising the manuscript.
Title: I would like it to be clearer in the title which of the psychometric properties were assessed.
Abstract: The Authors write: "The acceptability of the instrument was supported by low missing data (range = 0.00%- 1.14%) and low maximum endorsement (range = 0.00%-1.82%). Cronbach’s Alpha coefficients for each domain (0.78, 0.80) and high item-total correlations (range = 0.58-0.88) indicated good reliability." Cronbach’s alpha coefficient measures the internal consistency - one part of the reliability - but not reliability in total. Can you please revise the whole manuscript to be very specific about what reliability or validity you have looked at? Maybe follow COSMIN (COnsensus-based Standards for the selection of health Measurement INstruments) in the nomenclature to follow a uniform standard?
Abstract: The Authors write: "Construct validity was also good, with a Spearman’s rank correlation coefficient of 0.34." What construct was measured here?
Reporting Guidelines: I believe it is always advisable to follow a reporting guideline when writing an article. Was this done in this case? If not, please follow COSMIN (or a similar guideline). Please report all the contents given in COSMIN in the paper and give a filled copy in the appendix.
Open Science: Considering Open Science and Transparency, datasets should be published with the papers. Databases such as the Harvard Dataverse (https://dataverse.harvard.edu/) or similar are suitable for this purpose. I want to encourage the authors to do the same.
Transparency: If I have checked correctly, neither the original publication "The use of semistructured interviews to assess quality of life impacts for patients with uveal 318 melanoma", nor this one contains the PROM/questionnaire/assessment. This is, again, in the spirit of transparency, absolutely necessary, and the PROM-UM should be included in the supplement.
Methods: What about the other types of reliability/validity? Are they evaluated elsewhere? ... Please explain and/or include them in the paper.
Author Response
Thank you for your careful review. Responses to your specific comments are provided in the attached document.

Reviewer 2 Report
This paper describes a new questionnaire to be used with patients who have undergone brachytherapy for uveal melanoma. I think it should be published. in the attached file I notes some small typos. In addition, in Table 1 the factor loadings for each factor indicate the (unique) contribution of each item to each factor. However, there is a final column on the right labelled uniqueness that I do not understand. It is not mentioned in the text. What is does it mean?

Author Response

(The authors gave the same response as above.)

Reviewer 3 Report
The submitted manuscript analyzes a new instrument to measure how patients feel after treatment for eye cancer. The purpose of the paper is clear. I have a few comments:
1. Abstract: “Construct validity was also good.” Please be more specific about how construct validity has been assessed.
2. Abstract: “good reliability,” “good difficulty.” I do not think that the label “good” is scientific. Please consider using “satisfactory,” “acceptable,” etc.
3. Abstract: Please mention that discrimination and difficulty were computed based on an item response theory (IRT) model (i.e., graded response model).
4. The major weakness of the manuscript is the inconsistent use of statistical methods. That is, how the discrete Likert scale items were used in the analysis. In exploratory factor analysis (EFA), data is treated as continuous. In IRT, ordinal item scores are modeled. There is no reason for the different treatment in the analyses. You can apply EFA based on polychoric correlations (please see psych package R). Moreover, I do not see a reason why the orthogonal rotation method varimax is preferred over the oblique rotation method promax. Please always use the more general promax method because it allows for easier interpretation. Assuming that factors (i.e., “domains” according to your terminology) are uncorrelated does not make sense.
5. Ensure that the graded response model is fitted separately for the two scales. Otherwise, dimensionality would be confounded with discrimination. Fitting a unidimensional IRT model would contradict the EFA finding that there are two factors.
---
Author Response

(The authors gave the same response as above.)

Round 2
Reviewer 1 Report
The authors have responded well to my concerns. I wish all the best for the publication.
Reviewer 3 Report
I am satisfied with the revision.